# Gender Differences in Usage and Subjective Appreciation of an Online Cognitive Behavioral Therapy for Wildfire Evacuees: Descriptive Study

**DOI:** 10.3390/jcm11226649

**Published:** 2022-11-09

**Authors:** Émilie Binet, Marie-Christine Ouellet, Jessica Lebel, Vera Békés, Charles M. Morin, Geneviève Belleville

**Affiliations:** 1School of Psychology, Laval University, 2325 Rue de l’Université, Quebec, QC G1V 0A6, Canada; 2Ferkauf Graduate School, Yeshiva University, 1165 Morris Park Ave, The Bronx, New York, NY 10461, USA

**Keywords:** usage data, gender, online treatment, cognitive behavioral therapy, natural disaster

## Abstract

Background: Based on the most common psychological difficulties of the evacuees from the 2016 Fort McMurray wildfires in Alberta, Canada, a therapist-guided cognitive behavioral self-treatment was developed. This study aimed to explore how gender influences the usage and subjective appreciation of the RESILIENT online treatment. Methods: Our study included 81 English-speaking evacuees with significant posttraumatic symptoms, or with some posttraumatic symptoms accompanied by at least mild depression symptoms or subclinical insomnia, and who logged into the platform at least once. Various usage and subjective appreciation variables were analyzed, including number of completed sessions, number of logins, number of words per session, perceived efforts, perception of usefulness and intention to continue using the different strategies. Results: No difference was detected in most objective usage indicators. The number of words written in sessions 7 and 10 was significantly greater for women than for men. Regarding subjective appreciation, men had a greater perception of having put strong efforts in the cognitive restructuring strategy, while women reported in a greater proportion that they wanted to continue using physical exercise as a behavioral activation strategy. Conclusions: Our study offers a first look into how women and men use online treatments, and what their preferences are.

## 1. Introduction

Interest for self-administered therapies delivered online is on the rise. The current COVID pandemic has illustrated the relevance of this type of therapy and its value, particularly in times of high mental health needs but limited resources. Online mental health interventions offer many benefits, such as an increased accessibility and cost-efficiency, e.g., [1] and represent a promising way to provide timely and effective care to as many people as possible after a natural disaster [2]. Many studies have suggested that internet-based cognitive behavioral therapy is effective for treating anxiety disorders, posttraumatic stress disorder (PTSD) and depression [3,4,5,6]. Adherence to online treatments (e.g., completing more treatment modules) is associated with increased effectiveness [7,8], as is the frequency of use of cognitive behavioral therapy skills taught in these treatments [9]. This highlights the importance of paying attention to how participants use online treatments, in addition to whether these treatments are effective.

Many authors have suggested that gender plays a role in adherence to online psychological treatments, although the literature on how online treatments are used according to gender is inconsistent. In their systematic review, Beatty and Binnion [10] concluded that available preliminary evidence points to gender being a significant predictor of adherence, although half of the reviewed studies did not reach this conclusion. The large heterogeneity in studies that were included in the review in terms of intervention target and duration, sample size, and definition and measure of adherence warrants cautious interpretation of the results. Many studies suggested that female gender is a significant predictor of adherence for anxiety, stress and depression treatments [11,12,13,14,15]. However, one study evaluating an online treatment for depression found male gender to be a predictor of number of completed modules [16]. Other results suggest that men and women have a similar adherence to anxiety, depression and insomnia online treatments [17,18,19,20,21,22,23,24,25,26,27]. Taken together, these results underline the difficulty of drawing clear conclusions on the effect of gender on treatment usage and the importance of carrying out more studies to evaluate and better understand this effect.

The few studies that have specifically examined the effect of gender on treatment usage were mostly limited to the measurement of objective indicators of treatment use, i.e., number of logins and number of completed modules [7,23]. While these indicators provide valuable information, they do not appear to be sufficient to understand whether men and women use online treatments differently and which adaptations could prove helpful to enhance engagement. Many authors have suggested that user experience is an important therapeutic process to consider in the development of mental health technologies, e.g., [28,29]. A qualitative meta-synthesis on user experience of computerized therapy showed that participants placed a high importance on treatments that are sensitive to their clinical needs and preferences and that include personalized material [30]. Taking user views and perceptions into consideration appears critical to the efficacy of mental health treatments and adherence of participants [28,31], yet participants’ subjective appreciation of the content of online treatments (e.g., what they found useful and used the most) has rarely been examined. In order to determine if exploring whether gender-specific treatment adaptations offers clinical value, it is important to first understand if the use of treatment differs between men and women, and to what extent.

Our team has previously developed an online cognitive behavioral intervention (the RESILIENT platform) to address psychological issues in people evacuated from the 2016 Fort McMurray (AB, Canada) forest fires, the costliest disaster in the history of Canada at the time [32]. Following the event, many evacuees reported significant posttraumatic, depressive and insomnia symptoms [33,34,35]. A randomized controlled trial indicated that the treatment was more effective than a waitlist condition in terms of symptom improvement [36]. In another previously published study, we observed that men and women reacted differently to the treatment [37]. Men showed greater improvements in insomnia severity and in reducing self-blame posttraumatic cognitions than women, in line with other studies that highlighted that men and women may respond differently to treatment after a traumatic event [38,39,40]. However, it remains unclear if men and women used or interacted with the online treatment differently, thereby explaining different treatment responses.

The purpose of the present study was thus to examine potential gender differences in the use of an online cognitive behavioral therapy using a variety of objective and subjective indicators. This opportunity to explore how men and women interacted with the treatment could improve our understanding of potential gender-specific treatment mechanisms and ultimately allow us to tailor online psychological treatments to optimize efficacy for women and men. The main goal was to examine whether the usage and subjective appreciation of treatment strategies differed between men and women and to explore which indicators are worth further investigating. Our study aimed to answer the four following questions: (1) To what extent did women and men use our online treatment?; (2) Which treatment strategies seemed the most and the least helpful to each gender?; (3) Which strategies were men and women willing to continue using after the treatment?; and (4) How did the perception of the engagement in the therapeutic strategies differ between men and women?

## 2. Methods

### 2.1. Participants and Procedure

Participants were English-speaking adults who were evacuated from their homes during the 2016 Fort McMurray wildfires. They were initially recruited by random telephone sampling to participate to the first part of the larger study [13]. A phone survey was conducted from May to July 2017 with 1510 participants to evaluate the most prevalent psychological symptoms in order to create an adapted online psychological treatment. Following the survey, participants who were interested could participate in the longitudinal component of the study, which included four clinical assessments at 6-month intervals over two years (T1-T4). Validated questionnaires administered online assessed posttraumatic, depression and insomnia symptoms. Participants who reported significant posttraumatic symptoms (PTSD Checklist for DSM-5 [PCL-5] ≥ 23), or with some posttraumatic symptoms (PCL-5 ≥ 10) accompanied by at least mild depression (Patient Health Questionnaire Depression Scale [PHQ-9] score ≥ 5) or subclinical insomnia symptoms (Insomnia Severity Index [ISI] score ≥ 8) were invited to receive the online treatment. The treatment study included a randomized controlled trial design, and participants were separated into different groups which received treatment at different times over the 1.5-year period (pilot group, treatment group and waitlist group). Over the course of the study (November 2017–May 2019), 81 participants accepted the invitation to participate and logged into the treatment platform at least once. This includes the pilot group (*n* = 8), which received the treatment between November 2017 and May 2018, the treatment group (*n* = 32), which received the treatment between May and November 2018, and the waitlist group (*n* = 26), which received treatment between November 2018 and May 2019. Some participants with newly developed symptoms (*n* = 15) were included in the waitlist group and received the treatment between November 2018 and May 2019. All of these participants were included in the present study, regardless of the time at which they received treatment, in order to increase statistical power.

### 2.2. Treatment

The RESILIENT online intervention is a self-directed therapist-assisted treatment. It includes 12 sessions aimed at reducing PTSD, depression and insomnia symptoms. It incorporates several evidence-based cognitive behavioral interventions such as in vivo exposure, sleep restriction and imagery rehearsal therapy (IRT). A detailed list of session content and strategies is provided in Table 1. Access to a session was granted to the participant after the completion of the previous one, and the participant retained access to it afterwards. Each treatment session included reflection questions and practical exercises where participants could write their thoughts or answers. An important part of the treatment was the interactive tools. Participants were instructed to complete the sleep diary every morning, which provided them with an automated calculation of sleep efficiency and sleep window time recommendation each week. The diaphragmatic breathing tool enabled participants to track their progress with this exercise, for which a frequency of twice a day was recommended. The pleasant activities tool consisted of an activity planner to support behavioral activation. Participants were instructed to fill this every week, planning enjoyable activities for the following week. The in vivo exposure tool was used by participants to track their progress with this exercise, with the recommendation of conducting multiple exercises in a week. A cognitive restructuring tool and a problem-solving tool were also proposed to support cognitive therapy techniques and provide problem-solving skills. These could be used by the participant as needed. Table 1 presents the sessions at which the different tools were introduced to participants.

After completing a given session, participants had a contact with their assigned therapist using the modality of their choice (videoconferencing, telephone or email) to discuss treatment content, ask questions and troubleshoot problems with its application. Therapists provided clients with encouragement and evaluated adherence to the treatment strategies as well as the need for reference to specialized crisis services. It was instructed that these calls should last approximately 30 min. The therapists were graduate psychology students who were supervised by a licensed psychologist specialized in the treatment of PTSD (V.B.). Participants were given a period of six months to complete the 12 sessions, with the recommendation of completing one session per week.

### 2.3. Measures

*Sociodemographic information* included gender, age, marital status, ethnicity, ethnic origin, immigration status and membership in a First Nation. Gender identity was evaluated with the question “What is your gender?”. Participants could select one of the following answers: female, male, non-binary/third gender, prefer to self-describe (specify) and prefer not to say.

*Psychological symptom severity* was evaluated with validated self-reported questionnaires to determine the eligibility of participants and assess the association of pre-treatment symptom severity with gender. PTSD symptoms were evaluated with the *PTSD Checklist for DSM-5* (PCL-5; [41]), depression symptoms with the *Patient Health Questionnaire Depression Scale* (PHQ-9; [42]), and insomnia symptoms with the *Insomnia Severity Index* (ISI; [43]). In these questionnaires, higher scores indicate a greater severity of symptoms.

*Treatment usage* data related to how participants used the treatment platform. The number of completed sessions, number of logins, number of days between first and last login, mean number of days between logins, number of entries in each of the treatment tools (*sleep diary*, *diaphragmatic breathing*, *pleasant activities*, in vivo *exposure*, *cognitive restructuring* and *problem-solving*) were exported directly from the platform in an Excel sheet.

At session 7 (i.e., at mid-point), participants were asked to evaluate their level of perceived efforts in the five treatment strategies introduced so far: in vivo exposure, pleasant activities, sleep restriction and hygiene, diaphragmatic breathing and cognitive restructuring (presented as two specific strategies: *changing the interpretation of events* and *challenging cognitive distortions*). This self-reported questionnaire included a 5-point Likert scale (from 1 = a little to 5 = very much). An average was computed to represent the overall perceived effort of participants in the first half of treatment.

Text entered by participants, i.e., comments, reflections and answers to questions, was exported directly from the platform, and the number of words (by session and total) was calculated in an Excel sheet (the specific content was not analyzed). The number of words from a participant was available only for the sessions in which they had participated, thus resulting in variation of sample size due to attrition over the course of the sessions. The modality of contact with the therapist, the number of communications with the therapist, total duration of communications and mean duration of communications were also documented.

*Subjective appreciation data.* At session 12 (i.e., at the end of treatment), participants completed self-report questionnaires that were integrated in the platform to evaluate each of the individual treatment strategies in terms of: if they perceived them as useful (yes or no), if they wanted to continue using them (yes or no) and if they perceived that they put strong efforts into applying them (yes or no). Seventeen strategies were evaluated by participants: sleep hygiene, sleep restriction, pleasant activities, physical activity, social support optimization, diaphragmatic breathing, in vivo exposure, cognitive restructuring (interpretation of events), cognitive restructuring (cognitive distortions), mindfulness practice, self-compassion, active problem solving, imaginal exposure (writing), radical acceptance, nightmare rescripting, imaginal exposure (writing and reading), and values, goals and committed action. Table 1 presents the sessions in which the different strategies were included.

It should be noted that since participants presented with various levels of posttraumatic, depressive and insomnia symptoms, they were instructed not to use the parts of the intervention which did not apply to them. This may have impacted the results (e.g., a participant with low or no insomnia may have fewer entries in the sleep diary, a lower level of perceived efforts in the sleep diary tool and might have scored “no” to the item “I found it useful” for the sleep strategies).

### 2.4. Data Analysis

Descriptive data was computed separately for men and women. For continuous variables (e.g., number of completed sessions, number of words), means and standard deviations were computed, and for dichotomous variables, frequencies and percentages were computed (i.e., assessment of treatment strategies). *t*-tests were conducted to test for a difference in means between men and women for the variables on a continuous scale, and Chi-square tests of independence were conducted to test for a difference in frequencies for dichotomous variables in men and women when assumptions were met (i.e., sufficient number of participants per cell). Effect sizes were assessed by calculating Hedge’s g using means, standard deviations, and sample sizes. The impact of potentially confounding variables (age, marital status, ethnicity, level of education, membership in a First Nation and pre-treatment symptom severity) on significant results was evaluated to increase confidence that the observed effects on usage data was attributable to gender, using *t*-tests for continuous variables and Chi-square tests of independence for nominal variables (when assumptions were met). All analyses were run with SPSS Statistics version 25 using a significance threshold of 0.05.

## 3. Results

### 3.1. Sample Description

Table 2 presents the sociodemographic characteristics of the sample, which includes 81 participants who logged into the platform at least once. Participant age ranged from 19 to 71, with a mean age of 45.4 (SD = 11.6). All participants identified as either female (58/81; 71.6%) or male (23/81; 28.4%) gender. Most participants identified as White (71/81; 87.7%), were married or in a relationship (58/81; 71.6%) and had a postsecondary level of education (64/81; 79.0%). *t*-tests and Chi-square analyses revealed no significant differences between men and women in sociodemographic factors (i.e., age, marital status, level of education) and in symptom severity at pre-treatment (see Table 2). Individuals who did not identify as White (*n* = 9) or who identified as members of a First Nation (*n* = 4) were not represented in a sufficient number in our sample to test whether those characteristics were statistically associated with gender. However, descriptive data suggested that a greater proportion of women (94.7%) than of men (73.9%) identified as White. It also seemed that a greater number of men (7; 30.4%) than women (1; 1.7%) had an immigration status: four of them self-reported a European origin, two an Asian origin, one an African origin and one did not provide data for origin.

### 3.2. Impact of Gender on Treatment Usage

#### 3.2.1. Usage Data

Results initially showed that the mean number of days between first and last login was greater for men than women (115.48 days vs. 80.83 days; see Table 3). However, we found an outlier in the men subgroup (i.e., 370 days), which was over three standard deviations above the mean. When removed, the mean for men dropped to 103.91 days (SD = 63.38) and the difference was no longer significant (*t*_78_ = 1.576, *p* = 0.119). No significant gender differences were found in the number of completed sessions, number of logins, mean number of days between logins and number of entries in tool (total and for each tool). Men reported a higher level of perceived efforts at mid-treatment than women in the cognitive restructuring tool (4.08 vs. 3.38, range: 1–5). A higher level of perceived effort in the cognitive restructuring tool was also found in participants identifying with an ethnicity other than White (*t*_37_ = −2.820, *p* = 0.004) and in participants with an immigration status *t*_17.313_ = −2.896, *p* = 0.005). No significant gender differences were found in perceived efforts at mid-treatment overall and for the other tools.

Women’s preferred mean of communication with the therapist for feedback related to the sessions was by telephone (55.3%), followed by videoconference (34.2%) and email (10.5%). Men’s preferred mean of communication was videoconference (56.3%), followed by telephone (31.3%) and email (12.5%). No statistically significant difference was found between genders in mean of communication (*χ*^2^(2, *N* = 54) = 2.730, *p* = 0.255), nor in number of communications, total and mean communication duration (see Table 3).

#### 3.2.2. Number of Words

Results for mean number of words per session by gender are presented in Figure 1. *t*-tests showed that mean number of words per session was higher in women than in men in session 7 (*t*_44_ = 2.022, *p* =.049) and in session 10 (*t*_37.577_ = 2.602, *p* = 0.013). In session 7, women wrote a mean of 212 words while men wrote a mean of 136 words, which represents a medium effect size (*g* = 0.636, 95% CI [0.006–1.266]). In session 10, women wrote a mean of 1087 words while men wrote a mean of 390 words, which represents a medium effect size (*g* = 0.655, 95% CI [0.012–1.297]). No significant gender difference was found in other sessions and in the total number of words. Number of words was associated with level of education, ethnicity, and immigration status. Participants identifying as White wrote a greater number of words in sessions 7 (*t*_18.001_ = 2.384, *p* = 0.014) and 10 (*t*_50.844_ = 2.425, *p* = 0.009) and participants with an immigration status wrote a lower number of words in session 10 (*t*_44.342_ = 2.099, *p* = 0.021).

#### 3.2.3. Subjective Appreciation of Treatment Strategies

Table 4 presents the number of men and women who perceived each of the 17 treatment strategies as useful. The strategies perceived as most useful by women were pleasant activities and diaphragmatic breathing (22/27; 81.5%) followed by values, goals and committed actions (21/27; 77.8%) and cognitive restructuring (*cognitive distortions*) (20/27; 74.1%). The strategies perceived as most useful by men were sleep hygiene, pleasant activities, social support optimization, cognitive restructuring (*interpretation of events* and *cognitive distortions*), mindfulness practice and self-compassion (11/15; 73.3%). Imagery rehearsal therapy, suggested for participants with posttraumatic nightmares, was the strategy perceived as least useful by men and women (respectively 2/15; 13.3% and 4/27; 14.8%). Chi-squared tests did not reveal any statistically significant difference in men and women’s perception of the strategies as useful, however many analyses could not be run or may have lacked statistical power due to insufficient number of observations per cell.

The strategies that women most wanted to continue using (Table 5) were physical activity (20/27; 74.1%), followed by pleasant activities (19/27; 70.4%), social support optimization and cognitive restructuring (*cognitive distortions*) (18/27; 70.4%). The strategies that men most wanted to continue using were sleep hygiene, pleasant activities, social support optimization, mindfulness practice and values, goals and committed actions (8/15; 53.3% for all strategies). Imagery rehearsal therapy was the strategy that men and women least wanted to continue (respectively 1/15; 6.7% and 3/27; 11.1%). Chi-squared tests revealed that a greater proportion of women (20/27; 74.1%) than men (6/15; 40.0%) wanted to continue using physical exercise after the intervention (*χ*^2^(1, *N* = 42) = 4.747, *p* = 0.029).

Strategies in which participants perceived they had put strong efforts are presented in Table 6. Women reported that they had put strong efforts into pleasant activities (16/27; 59.3%), cognitive restructuring and self-compassion (13/27; 48.1%). Men reported that they put strong efforts into cognitive restructuring (cognitive distortions) (10/15; 66.7%), pleasant activities and cognitive restructuring (8/15; 53.3%). Radical acceptance was the strategy for which men perceived they had put the lowest level of efforts into (0/15; 0.0%), while for women, it was imagery rehearsal therapy (1/27; 3.7%). Chi-squared tests revealed that a greater proportion of men (10/15; 66.7%) than women (9/27; 33.3%) perceived they had put strong efforts into the cognitive restructuring (*cognitive distortions*) strategy (*χ*^2^(1, *N* = 42) = 4.325, *p* = 0.038).

Chi-square tests of independence revealed that age, marital status, ethnicity, level of education, and membership in a First Nation were not associated with these variables (strategy perceived as useful, intention to continue, perceived strong efforts; all *p*s < 0.05), but some Chi-square tests were inconclusive because of insufficient number of observations per cell.

## 4. Discussion

The goal of our study was to explore and describe gender differences in the usage data and subjective appreciation of an online cognitive behavioral treatment for natural disaster evacuees with posttraumatic, depression and insomnia symptoms. Results showed that the usage of the treatment platform seemed mostly similar for men and women (e.g., objective indicators such as number of logins and number of completed sessions), but gender differences emerged in the number of words written in the platform, and in the subjective appreciation and perceived level of efforts put in applying some treatment strategies. To our knowledge, such gender differences have not been previously reported in clinical studies of online cognitive behavioral treatments. It is therefore challenging to tie our findings to the specific literature on gender differences in the use of online treatments. In the next section, we expand on the observed differences by generating tentative explanations and relating our findings to the broader literature on gender differences in mental health and cognition. We also provide suggestions for further research.

Women wrote a greater number of words than men in sessions 7 and 10. Session 7 included self-assessment of symptoms and progress and consolidation of skills. Participants were invited to write down their experiences for five main treatment domains: posttraumatic symptoms, resilience, sleep, stress management and cognitive therapy. They were also invited to write down their goals for the next five sessions (general goal, specific goals and activities to achieve them), to review their exposure hierarchy (satisfaction with results, what helped them and what held them back) and write their plan for the next week regarding exposure. Session 10 included IRT for nightmares, and participants were invited to write their nightmare (if any) in detail, as well as their new dream script. Session 10 also included imaginal exposure to traumatic memories, and participants were invited to write down the narrative of their trauma, including as many vivid elements as possible (e.g., emotions, sensations and thoughts during the fires and evacuation). They were also invited to review their exposure exercises (progress, successes, difficulties, questions, plan for next weeks), the planning of pleasant activities (progress, inclusion of social and physical activities) and their sleep (improvements, compliance with sleep window, difficulties, plan for next weeks). A greater number of words for women than for men in these sessions suggest that women were more engaged in these parts of treatment; they may have felt a greater need to express themselves through writing or a natural tendency to expand and write about their experience in a more detailed manner. It may also suggest that these therapeutic strategies were more challenging for men.

The association between gender and number of words written in an online treatment has not been previously examined, but many gender differences have been reported in the study of language more generally. For instance, women tend to use a language that is more elaborate and affective than men, whose language is typically more direct and instrumental [44]. Gender differences favoring women were also noted in writing fluency tasks, where women told longer stories than men in autobiographic memory styles and recall of affective experiences [45,46,47]. These characteristics may make the writing about traumatic experience in detail easier or more accessible for women than men. Women are also typically more likely to use emotion-based coping strategies than men, e.g., [48,49] and thus, they might experience a greater ease than men in putting their emotions into words. When examining women and men’s subjective appreciation of the strategies from these sessions, they reported in a similar proportion that they perceived them as useful and that they wanted to continue to implement them in their lives. Thus, the length of written text in the platform might not be indicative of the perceived usefulness of the strategy. The impact of writing a greater or lower number of words in an imaginal exposure exercise is difficult to interpret in the context of a multi-strategy online treatment and, to our knowledge, has not been examined in the literature. Based on the rationale for this therapeutic strategy [50], one could suppose, on the one hand, that a lower number of words for men could indicate avoidance of some aspects of the traumatic event, perhaps emotional aspects. On the other hand, one could also make a case that a higher number of words for women reflects avoidance, as they may include unnecessary elements distracting them from the actual trauma narrative. A closer qualitative analysis of content would be necessary to further these questions.

Further, these results should be interpreted in the light of associations that were found between other sociodemographic variables and the number of words. Participants who were immigrants wrote a significantly lower number of words in some sessions, and there were more men than women with an immigration status in our sample. Consequently, immigration status might have been a confounding factor in the assessment of gender differences for this variable. It is plausible that English was not a first language for some immigrants, and that it influenced the length of their written accounts. Unfortunately, data on first language was not collected in this study, and, to our knowledge, the relationship between immigration status, native language and online treatment usage has not yet been examined in the literature. This factor appears important to consider in the development of treatments that require participants to write detailed responses. A Statistics Canada census has shown that in a literary task, a greater proportion of immigrants whose first language was different than the test language scored at the lowest level on a prose literacy scale, compared to immigrants whose first language was the test language and Canadian-born participants [51]. Future research should investigate the impact for a participant to receive an online treatment in a second language, and whether this influences treatment efficacy. Nonetheless, the development of a more inclusive approach to online treatments, offering content that is available in different languages, or offers alternative to written tasks (e.g., graphs, audio), seems of high importance.

Results revealed a significantly greater reported level of efforts put in cognitive restructuring at mid-treatment for men than women. This included the two cognitive restructuring strategies presented to participants: *changing the interpretation of events* and *challenging cognitive distortions*. At the end of treatment, men also reported in a significantly greater proportion than women that they had put strong efforts in the cognitive restructuring (*cognitive distortions*) strategy. Men appeared to be highly engaged in the cognitive restructuring strategy, both at mid-treatment and at the end of treatment, suggesting sustained effort over time, even when it was no longer the emphasis of treatment. These results are consistent with gender differences in the use of strategies to cope with stress. Men’s coping strategies are more likely to be based on problem-solving which seems consistent with cognitive restructuring, while women are more likely to use emotion-based coping to deal with stressors, e.g., [48,49].

Gender differences were also noted in the subjective appreciation of some strategies, although no difference was detected in most of them. Interestingly, the strategies perceived as most useful were not necessarily the ones that participants most wanted to continue or the ones in which participants perceived they had put a lot of effort into (e.g., sleep hygiene and self-compassion for men). The results also revealed that a significantly greater proportion of women wanted to continue using the physical exercise strategy. It appears that physical activity as a behavioral activation strategy resonated more with women in this sample, but the reasons for this effect are unclear. One hypothesis lies in the type of work evacuees did before the fires, with men presumably more likely to be involved in physical labor in the Fort McMurray area [52,53].

It is important to acknowledge that, with the exception of number of words in sessions 7 and 10, we did not detect a gender difference in usage data indicators, including number of completed modules and in number of logins, which are common indicators in treatment usage studies [7]. These results echoed the findings of multiple studies with populations suffering from anxiety, depression and insomnia disorders [18,21,24], which found no significant associations between gender and online program attrition, completion or adherence, i.e., number or completed modules. To our knowledge, only two studies targeted PTSD in their online treatment, and neither study detected a gender effect in attrition despite using large sample sizes [2,19]. The finding that suggests men and women used the platform similarly is not surprising, considering our previous finding that the improvement of men and women on most efficacy outcomes (e.g., decrease in posttraumatic and depressive symptoms) was similar [37]. A similar treatment usage and similar improvement in men and women is consistent with the association frequently reported between adherence to online treatments and increased effectiveness [7,8].

The present results should be interpreted with caution given the exploratory nature of the study. Our limited sample size suggests that our analyses may have lacked statistical power to detect differences. Another limitation is the fact that ethnicity and immigration status were not equally distributed among men and women in our sample. However, this reflects the socioeconomic context of the Fort McMurray area, where the oil sands of the region represent an important sector of employment, especially for men, and attracts workers from all over Canada and from outside the country [52,53]. Although our study was focused on the examination of gender differences, our assessment was limited to self-identified gender and did not measure adhesion to gender roles or biological sex. As the study of subjective appreciation variables was intended to be exploratory, the questionnaire used was not a validated measure. We believe, still, that its inclusion in the platform constitutes a methodological strength that allowed for a first step in studying gender differences beyond treatment efficacy.

Despite these limitations, our study is the first to explore how gender can influence usage data and subjective appreciation of an internet-based cognitive behavioral therapy. Our results expanded on the literature on gender differences in online treatments, which has been limited to a small number of objective indicators. Although many of the usage and subjective appreciation variables of our online cognitive–behavioral treatment were revealed to be similar for men and women, some differed significantly (i.e., number of words in some sessions, perceived level of efforts in cognitive restructuring) and are worth investigating further. This research avenue could provide important clinical insight into how existing and future online treatments could be improved to include gender-specific adaptations. Our study suggests that offering a variety of therapeutic strategy choices to men and women allows them to appreciate and use some more than others while reaching similar therapeutic gains, which seems promising. It highlights the relevance of exploring this line of research further, as the availability of online treatments will most definitely keep expanding. The question of how they should be adapted to sociodemographic and cultural factors is a crucial one, especially as our comprehension of sex and gender becomes more refined.

## Figures and Tables

**Figure 1 jcm-11-06649-f001:**
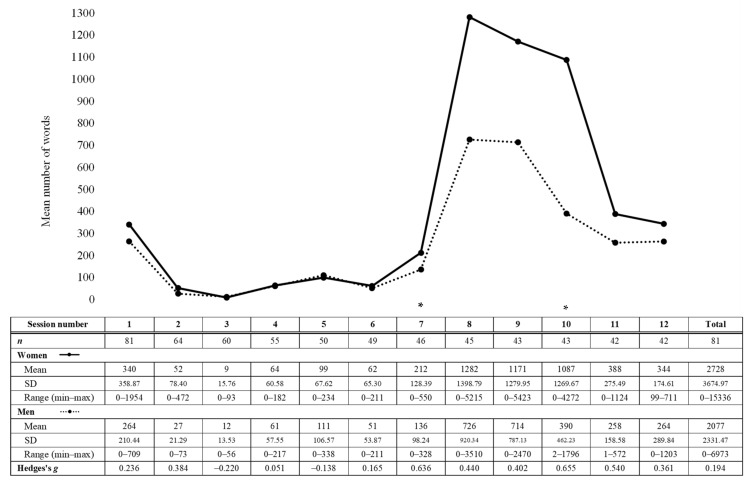
Means, standard deviations, range and effect sizes for number of words per session by gender. Note. Since participants presented with various levels of posttraumatic, depressive and insomnia symptoms, some of them did not use some parts of the intervention which did not apply to them. * *p* < 0.05.

**Table 1 jcm-11-06649-t001:** Session content, strategies covered and tools introduced.

Session	Session Content	Strategies Covered and Tools Introduced
(1) *Normal reactions to abnormal events*	▪ Overview of treatment ▪ Empowerment ▪ Posttraumatic stress disorder ▪ Resilience ▪ Sleep and insomnia ▪ Daily self-monitoring	▪ Psychoeducation ▪ Self-assessment of symptoms ▪ Daily self-monitoring ▪ Sleep hygiene ^a,b^ → **Tool introduced:** Sleep diary
(2) *Catching my breath*	▪ Phases of reactions to a natural disaster/traumatic event ▪ Diaphragmatic breathing ▪ Resilience and pleasant activities ▪ Sleep habits	▪ Psychoeducation ▪ Diaphragmatic breathing ^a,b^ ▪ Pleasant activities ^a,b^ ▪ Sleep hygiene ^b^ ▪ Sleep restriction ^a,b^ → **Tools introduced:** Diaphragmatic breathing tool Pleasant activities tool
(3) *Getting out there*	▪ Avoidance and in vivo exposure ▪ Role of physical activity in building resilience ▪ Sleep habits	▪ Psychoeducation ▪ In vivo exposure ^a,b^ ▪ Pleasant activities (physical exercise) ^b^ ▪ Review of previously covered strategies → **Tool introduced:** In vivo exposure
(4) *Thinking out loud*	▪ In vivo exposure ▪ Posttraumatic cognitions	▪ Psychoeducation ▪ ABC mode ▪ In vivo exposure ^b^ ▪ Cognitive restructuring ^a,b^ (changing interpretation of events) ▪ Review of previously covered strategies → **Tool introduced:** Cognitive restructuring tool
(5) *Paying attention to how I talk to myself*	▪ Cognitive distortions ▪ Unhelpful internal monologue	▪ Psychoeducation ▪ Cognitive restructuring (challenging cognitive distortions) ^b^ ▪ Review of previously covered strategies
(6) *Re-connecting with myself and others*	▪ Mindfulness ▪ Social support	▪ Psychoeducation ▪ Mindfulness meditation ^b^ ▪ Social support optimization ^b^ ▪ Review of previously covered strategies
(7) *My progress to date*	▪ Overview and assessment of progress ▪ Self-criticism and self-compassion	▪ Self-assessment of symptoms ▪ Psychoeducation ▪ Consolidation of skills ▪ Self-compassion ^b^ ▪ Review of previously covered strategies
(8) *Revisiting difficult memories*	▪ Posttraumatic memories and exposure ▪ Active problem solving ▪ Social support	▪ Psychoeducation ▪ Imaginal exposure to traumatic memories (writing) ^b^ ▪ Active problem solving ^b^ ▪ Social support optimization ^b^ ▪ Review of previously covered strategies → **Tool introduced:** Problem-solving tool
(9) *Keep moving forward*	▪ Radical acceptance of the past ▪ Posttraumatic memories and exposure	▪ Radical acceptance ^b^ ▪ Psychoeducation ▪ Imaginal exposure to traumatic memories (writing and reading) ^b^ ▪ Review of previously covered strategies
(10) *Taking control*	▪ Nightmares and imagery rehearsal therapy ▪ Posttraumatic memories and exposure	▪ Psychoeducation ▪ Imagery rehearsal therapy ^b^ (taking control of nightmares) ▪ Imaginal exposure to traumatic memories ▪ Review of previously covered strategies
(11) *Looking ahead*	▪ Resilience ▪ Values and life goals	▪ Reflection on one’s values, determination of goals and committed actions ^b^ ▪ Review of previously covered strategies
(12) *Preparing my toolkit for life*	▪ Triggers and other warning signs ▪ Review of progress	▪ Relapse prevention ▪ Self-assessment of symptoms ▪ Review of strategies toolbox

^a^ Strategies assessed at session 7; ^b^ strategies assessed at the end of session 12.

**Table 2 jcm-11-06649-t002:** Sociodemographic and clinical characteristics of participants.

Sample Characteristics	Total Sample	Men	Women	Group Differences
	(*N* = 81)	(*n* = 23)	(*n* = 58)	
	M (SD)	M (SD)	M (SD)	
**Age**	45.4 (11.6)	46.1 (13.5)	45.1 (10.9)	*t*_77_ = 0.345, *p* = 0.366
	*n* (%)	*n* (%)	*n* (%)	
**Gender**				
Female	58 (71.6)	-	-	-
Male	23 (28.4)	-	-	-
**Member of a First Nation**	4 (4.9)	1 (4.3)	3 (5.2)	-
**Ethnicity**				-
White	71 (87.7)	17 (73.9)	54 (93.1)	-
Other ^a^	9 (11.1)	6 (26.1)	3 (5.2)	-
**Immigration status**	8 (9.9)	7 (30.4)	1 (1.7)	-
**Marital status**				χ^2^_1_ = 0.700, *p* = 0.403
Single, separated, divorced or widowed	23 (28.4)	5 (21.7)	18 (31.0)	-
Married or partner	58 (71.6)	18 (78.3)	40 (69.0)	-
**Education**				χ^2^_1_ = 0.813, *p* = 0.367
Primary or Secondary	16 (19.8)	6 (26.1)	10 (17.2)	-
Postsecondary	64 (79.0)	17 (73.9)	47 (81.0)	-
**Clinical severity scores at pre-treatment**				
PCL-5	27.0 (14.8)	31.0 (15.6)	25.3 (15.8)	*t*_79_ = 1.474, *p* = 0.072
PHQ-9	10.9 (6.4)	11.7 (6.4)	10.7 (6.4)	*t*_79_ = 0.598, *p* = 0.276
ISI	16.9 (5.9)	17.2 (5.5)	16.7 (6.0)	*t*_79_ = 0.351, *p* = 0.363

Note: M = mean; SD = standard deviation. Totals did not always reach 100% since participants could choose not to answer (prefer not to say). ^a^ Including in decreasing proportion, Asian/Middle Eastern/Pacific Islander, Metis, Native North American or North American Indian, African (Northern, Eastern, Central, Western or Southern).

**Table 3 jcm-11-06649-t003:** Online treatment usage data for men and women (*N* = 81).

	Men (*n* = 23)	Women (*n* = 58)	*t*	*p* Value
	M (SD)	M (SD)		
Number of completed sessions	8.78 (4.61)	7.21 (4.89)	−1.329	0.188
Number of logins	21.91 (19.44)	20.90 (21.60)	−0.196	0.845
**Number of days between first and last login ** ^a^	**115.48 (83.14)**	**80.83 (56.59)**	**2.161**	**0.034**
Mean number of days between logins	9.62 (9.65)	7.61 (8.49)	0.871	0.387
Number of feedback communications with therapist	8.92 (3.53)	6.92 (4.94)	−1.540	0.136
Total duration of feedback communications with therapist (minutes)	293.63 (116.74)	258.84 (161.79)	−0.560	0.579
Mean duration of feedback communications with therapist (minutes)	30.42 (7.90)	30.98 (10.62)	0.178	0.859
Number of entries in tools—Total	29.48 (49.21)	23.47 (29.94)	0.671	0.504
Sleep diary	17.87 (25.19)	13.97 (19.42)	−0.748	0.457
Pleasant activities tool	4.74 (6.70)	4.76 (6.36)	0.012	0.990
Diaphragmatic breathing tool	2.87 (9.63)	1.66 (2.96)	−0.869	0.387
In vivo exposure tool	1.78 (6.86)	1.64 (4.11)	−0.117	0.907
Cognitive restructuring tool	1.17 (4.20)	0.95 (1.75)	−0.343	0.732
Problem solving tool	1.04 (3.54)	0.50 (1.11)	−1.055	0.295
Perceived level of efforts at mid-treatment (1–5)—Overall	3.67 (0.68)	3.44 (0.61)	1.135	0.263
In vivo exposure tool	3.15 (1.41)	2.80 (1.32)	0.791	0.434
Pleasant activities tool	4.00 (0.58)	4.00 (0.91)	0.000	1.000
Sleep diary	1.03 (0.29)	1.28 (0.23)	−0.312	0.756
Diaphragmatic breathing tool	3.85 (0.99)	3.53 (0.86)	1.048	0.301
**Cognitive restructuring tool**	**4.08 (0.90)**	**3.38 (0.90)**	**2.276**	**0.029**

Note: Significant results (*p* < 0.05) are identified with a bold font. Totals did not always reach 100% since participants could choose not to answer (prefer not to say). Since participants presented with various levels of posttraumatic, depressive and insomnia symptoms, some of them did not use some parts of the intervention which did not apply to them. ^a^ When removing one outlier in the men subgroup (i.e., 370 days), the mean for men drops to 103.91 days (SD = 63.38) and the difference was no longer significant (*t*_78_ = 1.576, *p* = 0.119).

**Table 4 jcm-11-06649-t004:** Subjective appreciation of treatment strategies by gender: perception of usefulness.

Treatment Strategies	Men (*n* = 15)	Women (*n* = 27)	*χ* _2_	*p* Value
Sleep hygiene	11 (73.3%)	16 (59.3%)	-	-
Sleep restriction	9 (60.0%)	12 (44.4%)	0.933	0.334
Pleasant activities	11 (73.3%)	22 (81.5%)	-	-
Physical exercise	10 (66.7%)	18 (66.7%)	0.000	1.000
Social support optimization	11 (73.3%)	19 (70.4%)	-	-
Diaphragmatic breathing	10 (66.7%)	22 (81.5%)	1.167	0.280
In vivo exposure	6 (40.0%)	13 (48.1%)	0.258	0.611
Cognitive restructuring (interpretation of events)	11 (73.3%)	18 (66.7%)	-	-
Cognitive restructuring (cognitive distortions)	11 (73.3%)	20 (74.1%)	-	-
Mindfulness meditation	11 (73.3%)	19 (70.4%)	-	-
Self-compassion	11 (73.3%)	15 (55.6%)	-	-
Active problem solving	9 (60.0%)	18 (66.7%)	0.187	0.666
Imaginal exposure (writing)	7 (46.7%)	16 (59.3%)	0.617	0.432
Radical acceptance	8 (53.3%)	14 (51.9%)	0.008	0.927
Imagery rehearsal therapy	2 (13.3%)	4 (14.8%)	-	-
Imaginal exposure (writing and reading)	7 (46.7%)	16 (59.3%)	0.617	0.895
Values, goals and committed actions	10 (66.7%)	21 (77.8%)	0.616	0.432

Note. Since participants presented with various levels of posttraumatic, depressive and insomnia symptoms, some of them did not use some parts of the intervention which did not apply to them.

**Table 5 jcm-11-06649-t005:** Subjective appreciation of treatment strategies by gender: intent to continue.

Treatment Strategies	Men (*n* = 15)	Women (*n* = 27)	χ^2^	*p* Value
Sleep hygiene	8 (53.3%)	17 (63.0%)	0.371	0.542
Sleep restriction	5 (33.3%)	12 (44.4%)	0.494	0.482
Pleasant activities	8 (53.3%)	19 (70.4%)	1.219	0.270
**Physical exercise**	**6 (40.0%)**	**20 (74.1%)**	**4.747**	**0.029**
Social support optimization	8 (53.3%)	18 (66.7%)	0.727	0.394
Diaphragmatic breathing	6 (40.0%)	14 (51.9%)	0.543	0.461
In vivo exposure	2 (13.3%)	7 (25.9%)	-	-
Cognitive restructuring (interpretation of events)	7 (46.7%)	14 (51.9%)	0.104	0.747
Cognitive restructuring (cognitive distortions)	7 (46.7%)	18 (66.7%)	1.601	0.206
Mindfulness meditation	8 (53.3%)	14 (51.9%)	0.008	0.927
Self-compassion	6 (40.0%)	12 (44.4%)	0.078	0.780
Active problem solving	7 (46.7%)	12 (44.4%)	0.019	0.890
Imaginal exposure (writing)	3 (20.0%)	7 (25.9%)	-	-
Radical acceptance	5 (33.3%)	9 (33.3%)	0.000	1.00
Imagery rehearsal therapy	1 (6.7%)	3 (11.1%)	-	-
Imaginal exposure (writing and reading)	4 (26.7%)	7 (25.9%)	-	-
Values, goals and committed actions	8 (53.3%)	17 (63.0%)	0.371	0.542

Note. Significant results (*p* < 0.05) are identified with a bold font. Since participants presented with various levels of posttraumatic, depressive and insomnia symptoms, some of them did not use some parts of the intervention which did not apply to them.

**Table 6 jcm-11-06649-t006:** Subjective appreciation of treatment strategies by gender: perception of having put strong efforts.

Treatment Strategies	Men (*n* = 15)	Women (*n* = 27)	χ^2^	*p* Value
Sleep hygiene	5 (33.3%)	10 (37.0%)	0.058	0.810
Sleep restriction	4 (26.7%)	8 (29.6%)	-	-
Pleasant activities	8 (53.3%)	16 (59.3%)	0.138	0.710
Physical exercise	4 (26.7%)	9 (33.3%)	-	-
Social support optimization	5 (33.3%)	9 (33.3%)	0.000	1.00
Diaphragmatic breathing	7 (46.7%)	12 (44.4%)	0.019	0.890
In vivo exposure	3 (20.0%)	8 (29.6%)	-	-
Cognitive restructuring (interpretation of events)	8 (53.3%)	13 (48.1%)	0.104	0.747
**Cognitive restructuring (cognitive distortions)**	**10 (66.7%)**	**9 (33.3%)**	**4.325**	**0.038**
Mindfulness meditation	5 (33.3%)	10 (37.0%)	0.058	0.810
Self-compassion	3 (20.0%)	13 (48.1%)	-	-
Active problem solving	6 (40.0%)	6 (22.2%)	1.493	0.222
Imaginal exposure (writing)	3 (20.0%)	11 (40.7%)	-	-
Radical acceptance	0 (0.0%)	8 (29.6%)	-	-
Imagery rehearsal therapy	2 (13.3%)	1 (3.7%)	-	-
Imaginal exposure (writing and reading)	1 (6.7%)	9 (33.3%)	-	-
Values, goals and committed actions	4 (26.7%)	12 (44.4%)	-	-

Note: Significant results (*p* < 0.05) are identified with a bold font. Since participants presented with various levels of posttraumatic, depressive and insomnia symptoms, some of them did not use some parts of the intervention which did not apply to them.

## Data Availability

The data presented in this study are available on request from the corresponding author. The data are not publicly available due to ethical restrictions.

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
