# Peer review of "Gender Differences in Usage and Subjective Appreciation of an Online Cognitive Behavioral Therapy for Wildfire Evacuees: Descriptive Study"

_jcm, 2022, doi:10.3390/jcm11226649_

Round 1

Reviewer 1 Report

Dear Authors, thank you for your work. I read your work with great interest, but it was really difficult to follow the content because there is no logical flow in the writing. For example, the introduction part is all mixed up with the introduction, methodology, and results, even the conclusion. You should revisit the write-up and create a logical flow so that it's easy for the reader to get important lessons from your study.

Here are examples from the introduction part which belongs to the methodology part,
"This study reports secondary data from a treatment study of people evacuated from the 2016 Fort McMurray (Alberta, Canada) forest fires".

"An online cognitive behavioral treatment (the RESILIENT platform) was developed by our research team to address these difficulties in evacuees and was proven to be effective in terms of symptom improvement 45 [11]."

Here are the examples of text, that belong to the Results part;

"We observed that men and women reacted differently to the treatment [12]. Men showed greater improvements in insomnia severity and in reducing self-blame posttraumatic cognitions than women, in line with other studies that highlighted that men and 48 women may respond differently to treatment after a traumatic event [13-15]."

These are just a few examples, the authors should revisit the whole manuscript for logical flow.

Below are the technical points, that the authors need to clarify to improve the manuscript, otherwise the findings of this research will not be replicable and applicable to others.

Why this study is based on a telephone survey that was conducted five years back in 2017. How you can justify that the analysis is applicable after a huge gap of five years?

The authors mentioned that "Over the course of the study (November 2017- May 2018), 81 participants accepted to participate and logged into the treatment platform at least once. All of these are included in the present study".  However, it is not mentioned, the data used in this study is based on which year? How many participants were allocated pilot group, how many were allocated treatment group, and how many were allocated, to the waitlist group? As this is not clear hence the rest of the analysis doesn't make any sense.  

Author Response

Responses to Reviewer 1’s Comments

  1. Comment: Dear Authors, thank you for your work. I read your work with great interest, but it was really difficult to follow the content because there is no logical flow in the writing. For example, the introduction part is all mixed up with the introduction, methodology, and results, even the conclusion. You should revisit the write-up and create a logical flow so that it's easy for the reader to get important lessons from your study.

Response: We wish to thank you for your review. Based on the comments, the flow of the introduction was improved in several ways:

  • We indicated the parts referring to our previous results to increase clarity (e.g., “Our previous analyses”) and grouped the different information regarding past research results (see lines 42-53)
  • We added the information about adherence to online treatments to the first paragraph, with a new justification as to why the subject is of importance (see lines 37-41)
  • We arranged the introduction to go from online treatments in general to our online treatment and our subject specifically (i.e., how gender related to adherence / treatment usage)

  1. Comment: Here are examples from the introduction part which belongs to the methodology part,
    "This study reports secondary data from a treatment study of people evacuated from the 2016 Fort McMurray (Alberta, Canada) forest fires".

    "An online cognitive behavioral treatment (the RESILIENT platform) was developed by our research team to address these difficulties in evacuees and was proven to be effective in terms of symptom improvement 45 [11]."

Here are the examples of text, that belong to the Results part;

"We observed that men and women reacted differently to the treatment [12]. Men showed greater improvements in insomnia severity and in reducing self-blame posttraumatic cognitions than women, in line with other studies that highlighted that men and 48 women may respond differently to treatment after a traumatic event [13-15]."

These are just a few examples, the authors should revisit the whole manuscript for logical flow.

Response: Although these extracts report on methods and results, they relate to previous studies conducted in our lab. As such, they  introduce our subject and research questions (i.e., gender differences in treatment usage, following our finding that men and women reacted to treatment in different ways). Such outlines are have been used in research articles on similar subjects (see examples below).

Batterham, P.J.; Neil, A.L.; Bennett, K.; Griffiths, K.M.; Christensen, H. Predictors of adherence among community users of a cognitive behavior therapy website. Patient Prefer Adherence 2008, 2, 97-105. PMID:19920949

Neil, A.L.; Batterham, P.; Christensen, H.; Bennett, K.; Griffiths, K.M. Predictors of adherence by adolescents to a cognitive behavior therapy website in school and community-based settings. J Med Internet Res 2009, 11(1), e6. doi:10.2196/jmir.1050

Price, M.; Gros, D.F.; McCauley, J.L.; Gros, K.S.; Ruggiero, K.J. Nonuse and dropout attrition for a web-based mental health intervention delivered in a post-disaster context. Psychiatry 2012, 75(3), 267-284. doi:10.1521/psyc.2012.75.3.267

3.      Comment: Below are the technical points, that the authors need to clarify to improve the manuscript, otherwise the findings of this research will not be replicable and applicable to others.

Why this study is based on a telephone survey that was conducted five years back in 2017. How you can justify that the analysis is applicable after a huge gap of five years?

Response: This study is based on the data following the 2016 Fort McMurray wildfires. The data specific to treatment usage was collected following the completion of the study (May 2019), and the data was analyzed at a later time. Participants received the treatment between November 2017 and May 2019. They were initially recruited by random telephone sampling to participate to the first part of the larger study, a phone survey to evaluate the most prevalent psychological symptoms in order to create an adapted online psychological treatment.

These clarifications were added to the Methods section (see lines 97-111)

  1. Comment: The authors mentioned that "Over the course of the study (November 2017- May 2018), 81 participants accepted to participate and logged into the treatment platform at least once. All of these are included in the present study". However, it is not mentioned, the data used in this study is based on which year? How many participants were allocated pilot group, how many were allocated treatment group, and how many were allocated, to the waitlist group? As this is not clear hence the rest of the analysis doesn't make any sense. 

Response: These clarifications were added to the Methods section (see lines 112-121)

Reviewer 2 Report

The subject of the manuscript is interesting, and the manuscript brings new insights and emphasizes the enormous potential of the online based treatments/interventions especially in the area of mental health that needs to be investigated further. The only objective limitation of the present research I see in the limited number of participants included. The study design I found appropriate, results are presented adequately, statistical analysis was suitable and the results, limitations of the study and the findings were discussed adequately.       

Author Response

Responses to Reviewer 2’s Comments

  1. Comment: The subject of the manuscript is interesting, and the manuscript brings new insights and emphasizes the enormous potential of the online based treatments/interventions especially in the area of mental health that needs to be investigated further. The only objective limitation of the present research I see in the limited number of participants included. The study design I found appropriate, results are presented adequately, statistical analysis was suitable and the results, limitations of the study and the findings were discussed adequately.

Response: We wish to thank you for your review.

Round 2

Reviewer 1 Report

Thank you for submitting the revised manuscript, but I feel the comments I made in the previous review are not really addressed. 

Author Response

Reviewer 1 responded to our revision (round 1) with the following comment: Thank you for submitting the revised manuscript, but I feel the comments I made in the previous review are not really addressed.” As such, we have revisited all of Reviewer 1’s previous comments carefully one by one and hope we have addressed them more clearly this time.

Previous Comment 1: Dear Authors, thank you for your work. I read your work with great interest, but it was really difficult to follow the content because there is no logical flow in the writing. For example, the introduction part is all mixed up with the introduction, methodology, and results, even the conclusion. You should revisit the write-up and create a logical flow so that it's easy for the reader to get important lessons from your study

Response: We acknowledge that some confusion may have been present because we present several of our previously published results. We have reorganized the introduction in the following ways to improve the flow:

  • We indicated the parts referring to our previous results to increase clarity (e.g., “Our previous analyses”, “previously developed”) and improved the links to our research objectives (lines 76-88)

“Our team has previously developed an online cognitive-behavioral intervention (the RESILIENT platform) to address psychological issues in people evacuated from the 2016 Fort McMurray (Alberta, Canada) forest fires, the costliest disaster in the history of Canada at the time [32]. Following the event, many evacuees reported significant posttraumatic, depressive and insomnia symptoms [33-35]. A randomized controlled trial indicated that the treatment was more effective than a waitlist condition in terms of symptom improvement [36]. In another previously published study, we observed that men and women reacted differently to the treatment [37]. Men showed greater improvements in insomnia severity and in reducing self-blame posttraumatic cognitions than women, in line with other studies that highlighted that men and women may respond differently to treatment after a traumatic event [38-40]. However, it remains unclear if men and women used or interacted with the online treatment differently thereby explaining different treatment responses.

  • We rearranged the introduction from general to specific, introducing the broader subject of gender differences in online CBTs, followed by our previous research and justification of our research objectives.

Previous Comment 2: Here are examples from the introduction part which belongs to the methodology part, "This study reports secondary data from a treatment study of people evacuated from the 2016 Fort McMurray (Alberta, Canada) forest fires".

Response: As to avoid any mention of the methods, this sentence is now completely modified for: “Our team has previously developed an online cognitive behavioral intervention (the RESILIENT platform) to address psychological issues in people evacuated from the 2016 Fort McMurray (Alberta, Canada) forest fires, the costliest disaster in the history of Canada at the time [10].” (line 76)

Previous Comment 2: "An online cognitive behavioral treatment (the RESILIENT platform) was developed by our research team to address these difficulties in evacuees and was proven to be effective in terms of symptom improvement 45 [11]."

Response: This sentence relates to previously published findings. We have made this clearer by adding that this was a randomized controlled trial “A randomized controlled trial indicated that the treatment was more effective than a waitlist condition in terms of symptom improvement [14].” (line 80)

Previous Comment 2: Here are the examples of text, that belong to the Results part;

"We observed that men and women reacted differently to the treatment [12]. Men showed greater improvements in insomnia severity and in reducing self-blame posttraumatic cognitions than women, in line with other studies that highlighted that men and 48 women may respond differently to treatment after a traumatic event [13-15]."

These are just a few examples, the authors should revisit the whole manuscript for logical flow.

Response: Again, this is previously published research. We made this clearer by adding “In another previously published study, we observed that men and women reacted differently to the treatment [15]. (line 82)

Previous Comment 3: Below are the technical points, that the authors need to clarify to improve the manuscript, otherwise the findings of this research will not be replicable and applicable to others.

Why this study is based on a telephone survey that was conducted five years back in 2017. How you can justify that the analysis is applicable after a huge gap of five years?

Response: We thank you for the previous comments, but the survey was not conducted in 2017. The survey was completed in 2019 and we are very confident the findings are still relevant today and will inform future research on gender differences in the usage of online cognitive-behavioral treatments. We acknowledge that some confusion may have been present since the project had different parts. The phone survey from 2017 had the objective to evaluate the most prevalent psychological difficulties in the evacuees to develop the online CBT. Then, participants who met certain criteria were invited to participate in the clinical treatment part of the study. The data collection ended in 2019. Clarifications were added to the Methods section (lines 103-118).

“Participants are English-speaking adults who were evacuated from their homes during the 2016 Fort McMurray wildfires. They were initially recruited by random telephone sampling to participate to the first part of the larger study [13]. A phone survey was conducted from May to July 2017 with 1510 participants to evaluate the most prevalent psychological symptoms in order to create an adapted online psychological treatment. Following the survey, participants who were interested could participate in the longitudinal component of the study, which included four clinical assessments at 6-months intervals over two years (T1-T4). Validated questionnaires administered online assessed posttraumatic, depression and insomnia symptoms. Participants who reported significant posttraumatic symptoms (PTSD Checklist for DSM-5 [PCL-5] ≥ 23), or with some posttraumatic symptoms (PCL-5 ≥ 10) accompanied by at least mild depression (Patient Health Questionnaire Depression Scale [PHQ-9] score ≥ 5) or subclinical insomnia symptoms (Insomnia Severity Index [ISI] score ≥ 8) were invited to receive the online treatment. The treatment study included a randomized controlled trial design, and participants were separated into different groups, which received treatment at different times over the 1.5-year period (pilot group, treatment group and waitlist group). Over the course of the study (November 2017- May 2019)”

Previous Comment 4: The authors mentioned that "Over the course of the study (November 2017- May 2018), 81 participants accepted to participate and logged into the treatment platform at least once. All of these are included in the present study".  However, it is not mentioned, the data used in this study is based on which year? How many participants were allocated pilot group, how many were allocated treatment group, and how many were allocated, to the waitlist group? As this is not clear hence the rest of the analysis doesn't make any sense. 

Response: Clarifications were added to the Methods section (lines 117-126).

Over the course of the study (November 2017- May 2019), 81 participants accepted to participate and logged into the treatment platform at least once. This includes the pilot group (n = 8), which received the treatment between November 2017 and May 2018, the treatment group (n = 32), which received the treatment between May and November 2018 and the waitlist group (n = 26), which received treatment between November 2018 and May 2019. Some participants with newly developed symptoms (n = 15) were included to the waitlist group and received the treatment between November 2018 and May 2019. All of these participants are included in the present study regardless of the time at which they received treatment, in order to increase statistical power.

Other changes to improve flow:

We also added information to the discussion section to improve flow and better justify the chosen structure and orientation of the discussion.

“To our knowledge, such gender differences have not been previously reported in clinical studies of online cognitive-behavioral treatments. It is therefore challenging to tie our findings to the specific literature on gender differences in the use of online treatments. In the next section, we expand on the observed differences by generating tentative explanations and relating our findings to the broader literature on gender differences in mental health and cognition. We also provide suggestions for further research. ” (lines 349-355)